# Recent Progress on Multifunctional Thermally Conductive Epoxy Composite

**DOI:** 10.3390/polym15132818

**Published:** 2023-06-26

**Authors:** Mei-Hui Zhou, Guang-Zhong Yin, Silvia González Prolongo, De-Yi Wang

**Affiliations:** 1Materials Science and Engineering Area, Escuela Superior de Ciencias Experimentales y Tecnología, Universidad Rey Juan Carlos, C/ Tulipán s/n, Móstoles, 28933 Madrid, Spain; m.zhou.2020@alumnos.urjc.es; 2Escuela Politécnica Superior, Universidad Francisco de Vitoria, Ctra. Pozuelo-Majadahonda Km 1, 800, Pozuelo de Alarcón, 28223 Madrid, Spain; amos.guangzhong@ufv.es; 3IMDEA Materials Institute, C/Eric Kandel 2, Getafe, 28906 Madrid, Spain; deyi.wang@imdea.org

**Keywords:** thermally conductive epoxy composites, electromagnetic interference shielding performance, flame retardancy, recyclability, electrothermal, toughness

## Abstract

In last years, the requirements for materials and devices have increased exponentially. Greater competitiveness; cost and weight reduction for structural materials; greater power density for electronic devices; higher design versatility; materials customizing and tailoring; lower energy consumption during the manufacturing, transport, and use; among others, are some of the most common market demands. A higher operational efficiency together with long service life claimed. Particularly, high thermally conductive in epoxy resins is an important requirement for numerous applications, including energy and electrical and electronic industry. Over time, these materials have evolved from traditional single-function to multifunctional materials to satisfy the increasing demands of applications. Considering the complex application contexts, this review aims to provide insight into the present state of the art and future challenges of thermally conductive epoxy composites with various functionalities. Firstly, the basic theory of thermally conductive epoxy composites is summarized. Secondly, the review provides a comprehensive description of five types of multifunctional thermally conductive epoxy composites, including their fabrication methods and specific behavior. Furthermore, the key technical problems are proposed, and the major challenges to developing multifunctional thermally conductive epoxy composites are presented. Ultimately, the purpose of this review is to provide guidance and inspiration for the development of multifunctional thermally conductive epoxy composites to meet the increasing demands of the next generation of materials.

## 1. Introduction

The miniaturization, integration, and functionality of electronic devices and the emergence of new applications such as energy transmission, 5G communication, and electronic packaging materials, have made heat dissipation an increasingly prominent challenge in many industries [1]. Epoxy resin (EP) is a widely used material in the electronic, electrical, and energy fields because of its excellent performance, including high tensile strength, thermal stability, excellent dimensional stability, electrical insulation, and chemical resistance. However, its low thermally conductive (TC) of approximately 0.2 W/(m·K) restricts its application in various fields, as it is primarily considered a heat insulator [2,3].

Mixing TC fillers directly with EPs has been established as a straightforward method for producing TC epoxy composites (ECs) [4,5]. However, increasing the filler content to enhance TC can result in reduced mechanical properties, processing difficulties, and increased costs, despite the improvement in TC [6,7]. To tackle these challenges, researchers are exploring two areas: modifying the molecular structure of the EP to increase its TC and searching for new filler structures that can provide enhanced TC. Some reports have shown promising results with ultra-high TC values [8] (>50 W/(m·K)), which is much higher than that of commercial use. Significant progress has been made in improving TC performance in the last several decades, and many reviews have recently been published on epoxy-based TC materials, covering topics such as EP structure design [9,10], filler options [11,12], and fabrication methods [13,14], etc. By selecting suitable EPs and fillers, TC with desired properties can be achieved.

In addition to improving TC performance, there are significant opportunities to develop multifunctional TC ECs [15,16,17]. For example, by introducing dynamic covalent bonds into TC ECs, it is possible to produce green degradable ECs with high TC [18,19]. Recent research has also yielded TC epoxy materials with electromagnetic interference (EMI) shielding [20], flame-retardant [21], electrothermal conversion [22], and tough properties [23]. These functionalities indicate that multifunctional thermally conductive ECs are critical for expanding their range of applications. The future trend for advanced TC epoxy materials involves designing them with multiple properties to increase their suitability for specific applications. However, there is currently a lack of comprehensive reviews that focus on multifunctional TC ECs.

This work provides a systematic overview of recent advancements in mainstream multifunctional TC ECs, by systematically discussing key scientific and technical issues to their various functionalities. In the first section, we provide an in-depth discussion of the definition, mechanisms, and research progress of TC EPs. The subsequent section presents recent progress on TC ECs with different functionalities (as shown in Figure 1). This section provides a concise introduction to the mechanisms, fabrication methods, and applications of TC ECs, as reported in the literature when discussing their performances.

First, we provide an overview of epoxy vitrimers with various dynamic bonds, which offer desirable properties such as recyclability, repairability, weldability, and malleability, and also highlight the major challenges associated with the development of TC epoxy vitrimers. Subsequently, TC ECs with EMI shielding functions are introduced. The EMI shielding mechanism of EMI shielding polymers is to absorb and scatter electromagnetic waves by the conductive particle’s connected conductive network and the polarization and capacitive effects of the polymer, thereby reducing the impact of electromagnetic radiation. We have studied the influence of the selection of conductive fillers on the TC and EMI shielding performance of the ECs. Concerning the flame-retardant function, the TC fillers and the flame retardant are discussed, as well as other methods for making the TC ECs with flame-retardant properties. Other multifunctional ECs widely study in the bibliography are thermos-electrical ones. There are many reported works based on the design of electro-thermal heating systems. The heating is determined by the Joule effect of the current flowing through the electrically conductive materials [24]. Whereas how to achieve a superb Joule effect with different conductive fillers has become a major problem. Therefore, in this section, we focus on the reasonable choice of electrically conductive filler type in antifogging and deicing for outdoor systems applications. In these applications, a highly electrically and thermally conductive is desired to be a more efficient material. Next, high TC with toughness is briefly discussed in ECs. Finally, we look forward to TC ECs future development trajectory and challenges, which will guide developing, designing, and implementing multi-purpose TC ECs.

**Figure 1 polymers-15-02818-f001:**
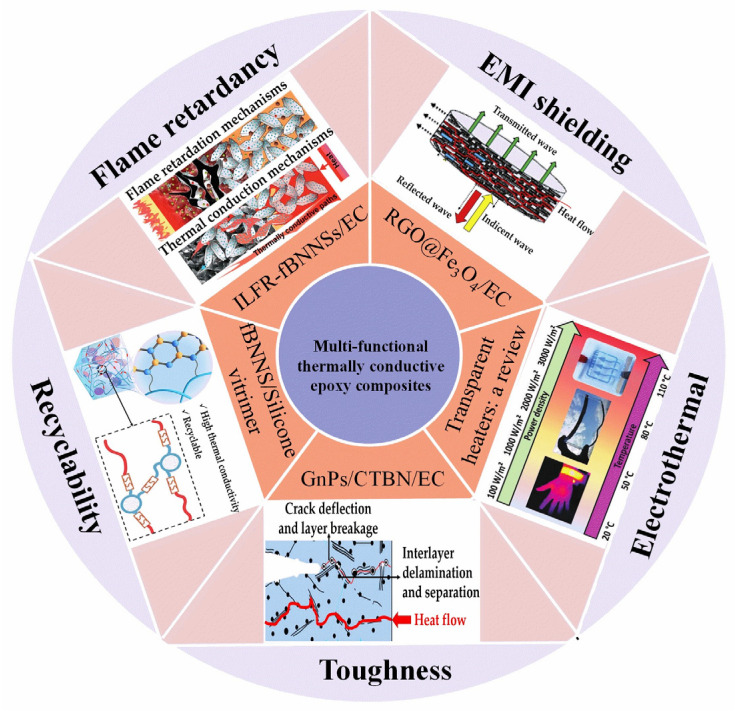
Brief introduction of thermally conductive epoxy resin with multifunctionalities. “EMI shielding”, reproduced with permission [25]. Copyright 2019, Reproduced with permission from Elsevier Ltd. “Flame retardant”, reproduced with permission [15]. Copyright 2018, Reproduced with permission from the Royal Society of Chemistry. “Recyclability”, reproduced with permission [26]. Copyright 2022, Reproduced with permission from Elsevier Ltd. “Electrothermal”, reproduced with permission [27]. Copyright 2020, Reproduced with permission from Wiley-VCH GmbH. “Toughness”, reproduced with permission [28], Copyright 2016, Reproduced with permission from Elsevier Ltd.

## 2. Thermally Conductive of Epoxy Resins: Definition, Mechanisms, and Parameters

### 2.1. Definition of Thermally Conductive

Thermally conductive refers to the ability of a material to conduct or transfer heat. In solids, heat can be transferred through phonons, which are energy quanta of atomic lattice vibrations, or charge carriers like electrons and holes in materials [29]. Highly crystalline materials, including graphene, metals, and ceramics, exhibit high TC [30]. Heat is transferred via two mechanisms in a crystal lattice: lattice vibration (phonons) and electron motion (electrons). In the lattice vibration mechanism, vibrational energy is transferred from one atom to its neighboring atoms, resulting in heat transport. In the electron motion mechanism, energy is transferred from high-energy electrons to low-energy electrons, leading to heat transport. The presence of a large number of free electrons in metals facilitates this process, as they can move through the lattice and transfer energy to neighboring atoms [31].

The Fourier equation describes the fundamental principle of heat conduction, considering the variation of temperature distribution concerning time and space. It expresses the relationship between temperature gradient and heat flux density [32], as shown in Figure 2, where heat flux density represents the rate of heat transfer per unit area. The equations are given by (1) and (2):(1)$$q=kA\frac{\Delta T}{L}$$
or
(2)$$J=\frac{q}{A}=-k\frac{dT}{dx}$$
where q is rate of heat transfer (W); J is heat flux (W/m^2^), k is TC W/(m·K); A is cross sectional transfer area (m^2^); ΔT is temperature different (°C); L is conduction path length (m).

It should be noted that the Fourier equation typically assumes materials to have linear, isotropic, and steady-state (or transient) properties when describing heat conduction. This means that the thermal conductivity of the material is the same in different directions and does not vary with time.

However, for certain special cases or complex materials, the Fourier equation may require modifications or further model development to account for nonlinear, anisotropic, or non-steady-state heat conduction properties. Such modified models may need to consider more complex physical processes, such as interfacial thermal resistance, lattice distortion, microscale defects, or phase transitions.

Amorphous polymers are saturated, there are no free electrons in them, and molecular motion is difficult [10]. In amorphous polymers, phonons are the primary carriers of heat energy. The mechanism of TC is significantly different for amorphous polymers compared to crystalline materials. The TC of amorphous polymers is hindered by various issues, including their highly coiled and entangled intrachain structure, loose chain packing with voids, and weak nonbonding interchain interactions, including van der Waals force. These factors limit the efficient transport of heat through the material [11]. To illustrate the difference in TC between amorphous polymers and crystalline materials, a Newton pendulum analogy can be used [32]. A crystalline substance is represented by an ordered pendulum and an amorphous polymer by a disordered pendulum. Figure 3 shows that when the ordered pendulum is vibrated (Figure 3a), the initial energy quickly propagates to the opposite end, while in the disordered pendulum (Figure 3b), the initial energy is mainly confined to the individual balls, causing localized vibrations. This analogy highlights how the lack of order and defects in amorphous polymers can impede heat transfer, leading to lower thermal conductivity compared to crystalline materials.

### 2.2. Thermally Conductive in Epoxy Resins

EPs were first discovered by Prileschajew in 1909 [33]. They are characterized by the presence of an epoxy or oxirane ring, which is a three-membered ring with an oxygen atom bonded to two carbon atoms as illustrated in Figure 4 [34]:

Since EPs are thermosetting materials, they can interact with a variety of curing agents, including amines, anhydrides, carboxylic acids, alcohols, and thiols [35]. These pre-polymers develop into a three-dimensional network with limitless molecular weight after curing, due to the functionality of epoxy monomer or/and hardener must be higher than 2.

The TC of EPs has been investigated using both equilibrium molecular dynamics (EMD) and nonequilibrium molecular dynamics (NEMD) methods. Varshney et al. [36] reported a TC value between 0.30–0.31 W/(m·K) at a temperature of 300 K, while Kline et al. [37] observed a range of TC values from 0.23–0.27 W/(m·K) as the temperature rose from 275 to 375 K. The low TC of most EPs is well-established theoretically and experimentally, and it is attributed to the presence of numerous defects and randomly oriented structures in the polymer chains. These factors create a tortuous path for the propagation of vibrational waves, which seriously affect phonon transmission and promote phonon scattering.

To overcome this limitation, researchers have attempted to develop EPs with high TC by introducing highly ordered structures, commonly called liquid crystal epoxy resins (LCER). This is accomplished by introducing mesogenic groups into the molecular chains of the epoxy. For example, Takezawa [38] synthesized liquid crystalline epoxy monomers with diphenyl benzoate groups, resulting in an ordered molecular structure with a cured epoxy TC exceeding 0.90 W/(m·K). Islam et al. [39] replaced the traditional amine crosslinker (DDS) with a cationic initiator (N-benzyl pyraziniumm hexafluoroantimonate, BPH) to achieve LCERs with a TC value ~141% higher (0.48 W/(m·K)) than that of amorphous amine-cured LCERs. The self-arranging capability of mesogenic groups into a highly ordered lattice structure substantially enhances the phonon transport of the epoxy.

Table 1 provides a summary of the thermal conductivities of various commercial EPs, which range from 0.19 to 0.96 W/(m·K). Amorphous EPs have much lower TC than LCERs.

### 2.3. Thermally Conductive Epoxy Composites

The TC mechanism of ECs is more complex compared to that of EPs. The TC value of ECs is dependent on various factors, including the characteristics of the TC fillers such as loading, shape, size, and surface morphology [1,30,45,46]. Moreover, the characteristics of the epoxy matrix, such as chain structures, crystal degree, and intermolecular interaction, also play a vital role in determining the TC value [10,11]. The functionalization of fillers and the final nano- and micro-structure of composites, such as dispersion degree and alignment, are other critical factors in achieving higher TC values [5,12,32,47]. In addition, the interface thermal resistance (ITR) between filler/filler, polymer/polymer, and filler/polymer also influences the TC of composites [48]. A systematic review of the relationship between TC value and various factors has been carried out in several reviews [1,30,49].

There are three common types of thermally conductive fillers: metal-based fillers (copper [50], silver [48], aluminum [51], carbon-based fillers (carbon nanotubes [52], carbon fiber [48], graphene [30], graphite [52]), and ceramic fillers (boron nitride [11], silicon carbide [53], aluminum oxide [11]). Figure 5 shows the TC of epoxy composites containing different types of fillers. EC with metallic-based fillers and carbon-based fillers are mainly used in heat transfer and dissipation areas where electrical insulation is not required, such as heat exchangers [52]. Thermally conductive Ceramic filled ECs are widely used in areas needing electrical insulation, such as printed circuit boards [53]. Different types of thermally conductive fillers have different characteristics and thermal conductivity. In practical applications, a comprehensive evaluation and selection of suitable thermally conductive fillers are required based on the required thermal conductivity targets, cost, processability, and other factors.

The simulation models can be used to predict the TC value of composites, analyze the effect of various factors on the TC, and describe the complex heat transfer mechanism in the composites. At the microscopic level, the EMD and NEMD are common simulation methods for composites. NEMD is more commonly used for TC calculation of composite materials because it can take into account any shape and structure of composite materials without making any assumptions or simplifications, and can describe the vibration motion of the phonons in detail [54]. For instance, Zhao et al. [55] used NEMD to investigate the relationship between TC and epoxy structure. The simulation results showed that the introduction of crosslinking bonds significantly contributes to higher TC in epoxy materials. From a macroscopic perspective, the finite element method (FEM) and finite difference method (FDM) are well-known methods. Saini et al. [56] utilized FEM to explore the impact of filler shape, size, distribution, and volume fraction on the TC value of silica-epoxy and alumina-epoxy composites. The simulation results suggested that the TC pathway can be built more efficiently in the direction of heat flow with random space distribution at a higher volume fraction.

The thermal conduction path theory is the most commonly accepted mechanism for explaining TC in ECs. This theory proposes that paths are formed through the contact of TC fillers within the epoxy matrix. Heat flux is then transferred along these TC filler paths or networks with lower thermal resistance. At low loading levels of TC fillers, they are isolated from each other and the epoxy matrix, resulting in a “sea-island” system with a low TC value and no significant improvement in the TC values of the ECs. However, as the loading of TC fillers increases, they begin to contact each other and form thermal conduction paths or networks. However, the TC does not require a percolation threshold like the electrical conductivity of polymer composites. No sharp increase in their TC is ever observed as a function of filler content. Unfortunately, simple mixing of the TC fillers and EP often results in random filling/epoxy interface distribution, resulting in high ITR and restricted TC values. Consequently, the primary strategies for fabricating high TC ECs are to construct an efficient heat transfer path and to decrease the ITR between the filler/epoxy.

## 3. Thermally Conductive Epoxy Composites with Multifunctional Properties

### 3.1. Thermally Conductive Epoxy Vitrimers

Traditional EPs are insoluble and infusible, which poses challenges for monitoring and repairing microcracks, and makes it difficult to reprocess, recycle, reshape, or reconfigure them [57]. Consequently, most aged, damaged, and discarded resins and composites are currently disposed of through incineration or are accumulated in landfills, resulting in severe environmental pollution and resource waste [58].

In 2011, a transesterification catalyst was added to an epoxy/acid or epoxy anhydride polyester network, which is a crosslinked network comprising a dynamic covalent bond, to create the first vitrimer [59]. In the case of exchangeable reactions, the crosslink density is not changed, and the vitrimer has excellent thermal and mechanical performance at service temperature. Exchangeable reactions (for example, transesterification) occur quickly when the temperature exceeds the topological freezing transition temperature (T_v_). Under external force, transesterification induces topological rearrangement and quick stress relaxation. This results in the epoxy vitrimers having the ability to flow like a viscoelastic fluid, making them suitable for reprocessing, reshaping, and recycling at high temperatures [60]. Through thermally-triggered and catalyst-accelerated transesterification, it is possible to modify the network topology of epoxy vitrimer. Several exchangeable reactions have also been investigated for epoxy vitrimers, including transesterification [19,59], disulfide exchange [61], imine bonds [62], and combinations of multiple dynamic bonds [63], as shown in Figure 6.

Doping has been used to improve the TC of materials since near epoxy vitrimers do not meet industrial requirements. For example, introducing micron boron nitride (mBN) into a transesterification-based thiol-epoxide vitrimer to prepare the highly TC, self-healing, and recoverable mBN/thiol-epoxy elastomer composites by hot pressing method. The TC of the composite with 60 wt% mBN was about four times higher than the pristine epoxy vitrimer, as shown in Figure 7a [64]. The uniform dispersion of fillers in vitrimers is the key point to producing composites with optimal performances. Surface modification of filler is a feasible way to improve the dispersion in the matrix. Weng et al. [26] modified BN and introduced the functionalized boron nitride nanosheets (fBNNS) into silicone vitrimer through a curing reaction as shown in Figure 7b, excellent TC of 1.41 W/(m·K) was obtained with 66 wt% fBNNS, which was 6 times higher than that of silicone vitrimer. However, when fBNNS was added 66 wt%, the recovery efficiency of silicone vitrimer decreased to more than 92.0% from the highest to 98.8%. It is worth mentioning that for most epoxy vitrimer composites, there is a trade-off between their enhanced properties and hindered topological rearrangement because the fillers restrict the mobility of polymer chains. High-loading fillers may impact the topology rearrangement while low-loading fillers would not be enough to enhance the TC value of epoxy vitrimer composites.

Besides the direct addition of fillers in vitrimers, Ding et al. [65] made epoxy vitrimer composites with selectively distributed multi-walled carbon nanotubes (MWCNTs) and hexagonal boron nitride (h-BN) for good TC and electrical insulation. To attain such a structure, MWCNTs were first added into the vitrimer precursors, then the cured MWCNTs/vitrimer composites were grounded into particles and mixed with the h-BN, the composites were finally successfully fabricated through compression molding, as shown in Figure 7c. The TC of 1 wt% MWCNTs/8 wt% h-BN/epoxy vitrimer composite (R0) reaches 0.83 W/(m·K). However, after being reprocessed under identical experimental conditions, the TC of the reprocessed 1 wt% MWCNTs/8 wt% h-BN/epoxy vitrimer composite (R1) decreased by 20% compared to 1 wt% MWCNTs/8 wt% h-BN/epoxy vitrimer composite. After the third retreatment (R3), the decrease in elongation at break and TC is more pronounced. This may be due to the aggregation of h-BN within the matrix and the disruption of TC pathways. Qi et al. [66] reported enhancing the interfacial interaction between fillers and epoxy vitrimer via polydopamine (PDA) coating to improve TC and strength. Interestingly, the coordination interactions formed by the catechol groups of PDA and Zn^2+^ (from the transesterification catalyst) also accelerated the stress relaxation of fillers/vitrimer composites.

### 3.2. EMI Shielding Thermally Conductive Epoxy Composites

The rapid development of electronic equipment and wireless communication technologies has led to increasing demand for highly integrated and fast wireless communication devices. However, these devices often generate substantial amounts of heat and EMI, which can seriously affect their normal operation and pose a threat to living organisms [67]. As a result, there is a growing interest in developing highly TC EPs with excellent EMI shielding efficiency for a variety of applications, including high-power electronics, wireless communications, portable devices, and self-driving automobiles.

The development of EMI shielding EPs with high TC is critical for effective shielding against unwanted electromagnetic waves and heat dissipation. EMI shielding requires appropriate materials to attenuate or block the transmission of electromagnetic waves, protecting the object from interference in space. Figure 8 illustrates the three mechanisms by which EMI attenuation can occur: reflection, absorption, and multiple reflections. When the electromagnetic waves come close to the surface of the shielding material, a part of the wave will be reflected by the resistance of the material, and it will be reduced exponentially. When the waves reach another surface of the shielding material, some of them will be re-reflected (multiple internal reflections), and the remaining waves will be transmitted [68].

EPs are widely used as insulating materials, but they provide poor EMI protection and low TC [69]. To address these limitations, researchers are exploring the addition of conductive fillers to enhance TC and EMI shielding. Various fillers, including carbon-based, metallic, and ceramic fillers, have been studied [70]. Carbon-based fillers, such as graphite, carbon nanotubes, and graphene, are favored for their exceptional electrical conductivity and TC properties [71,72].

Bao et al. [73] successfully prepared 3D sulfanilamide-modified expanded graphite/epoxy (EG-SA/EP) composites by a pre-filling and hot pressing method. The high TC (98 W/(m·K)), the EMI shielding (85 dB), and the electrical conductivity (7153 S/m) of the composites with 70 wt% EG-SA were demonstrated. Hybrid-reinforced epoxy composites, which combine multiple fillers, can offer even greater benefits. For example, Liu et al. [25] fabricated magnetic reduced graphene oxide (RGO)@Fe_3_O_4_ nanoplatelets (NPs) through electrostatic self-assembly and co-precipitation technique (Figure 9a). The resulting nanocomposites were aligned during curing using external magnetic fields and exhibited outstanding TC (1.213 W/(m·K)) and EMI shielding of 13.45 dB at 8.2 GHz. Guo et al. [74] created MWCNT-Fe_3_O_4_@Ag nanoparticles, which were then incorporated into an epoxy matrix via a blending-casting method to form composites with 15 wt% MF-10 (Figure 9b). The resulting composites exhibited satisfactory EMI shielding (35 dB), electrical conductivity (0.280 S/cm), and TC (0.46 W/(m·K)). Wu et al. [20] prepared multifunctional ECs with interconnected carbon nanotubes (CNTs) and carbon fibers cloth (CFC), grown in situ on the surface of CFs from a 2D bimetallic leaf-shaped Co/Zn zeolitic imidazolate framework structure, as shown in Figure 9c. The CFC@CNTs/EP exhibited an integrated in-plane TC of 7.50 W/(m·K), and through-plane TC of 1.96 W/(m·K), and an EMI shielding effectiveness of 38.4 dB. Yang et al. [75] prepared a novel EC using graphite nanosheet (GNP) and partially de-crosslinked waste epoxy resin powder(WEP). The introduction of partially de-crosslinked WEP powder in epoxy via a facile solution mixing and hot-pressing process helps to form an effective three-dimensional GNP network (Figure 9d), leading to a maximum TC of 10.1 W/(m·K) and an exceptionally high EMI shielding efficiency of 106.3 dB.

Electronic packaging materials play a crucial role in preventing short circuits and maintaining circuit stability, and their electrical insulation performance is critical to achieving this. However, the achievement of superior electrical insulation properties does not increase the EMI shielding performance. To address this challenge, researchers have concentrated on developing materials that possess both TC and EMI shielding performance while retaining electrical insulation properties. Zhang et al. [76] developed a sandwich network structure with an insulation layer and an EMI shielding intermediate layer, which showed through-plane and in-plane TC values of 3.051 W/(m·K) and 3.365 W/(m·K), respectively. The material also demonstrated excellent electrical insulation (6.61 × 10^12^ Ω·cm) and excellent EMI shielding (>27.68 dB). Similarly, Zhang et al. [77] prepared a layered structure low melting-point alloy (LMPA)/EC by self-sedimentation of high-density LMPA. The metal characteristic of LMPA imparted good TC and excellent EMI shielding performance to the epoxy matrix, while the good compatibility of LMPA with the EP enabled overall electrical insulation. The composite attained a TC value of 1.23 W/(m·K) on the LMPA side, a total EMI shielding value of 35.56 dB at 30 GHz, and an electrically conductive of 5 × 10^−8^ S/cm with a loading of 20 vol% LMPA.

### 3.3. Flame-Retardant Thermally Conductive Epoxy Composites

When exposed to heat or flames, EPs undergo pyrolysis, a process in which the resin molecules break down into smaller fragments [78]. This can lead to the release of flammable gases or volatile organic compounds (VOCs), which can contribute to flame propagation and the generation of smoke and toxic fumes [79,80]. In the electronics and automotive industries, where heat dissipation is critical, TC EPs are used. However, these resins must meet safety regulations or minimize the risk of fire, which necessitates a certain level of FR [81]. Hence, it is imperative to develop an appropriate approach to fabricate high-performance ECs with high TC and FR that would be essential for both academic investigations and industrial applications.

Typically, the most efficient and straightforward approach to enhance the properties of composites is by incorporating inorganic fillers or additives into the epoxy matrix. For instance, Li et al. [15] demonstrated that the addition of non-covalent ionic liquid-modified BN, organic-inorganic hybrid fillers, can considerably increase the FR and TC of ECs. Due to their outstanding qualities, several 2D materials, including h-BN [82], graphene [83], black phosphorus [84], MXene [85], and molybdenum disulfide (MoS_2_) [86], have also been extensively reported as effective TC or flame-retardant fillers, owing to their exceptional properties.

Despite the excellent barrier properties and thermal stability of 2D materials, their addition alone is not adequate to meet FR rating requirements, as they lack effective flame-retardant ingredients. As a solution, numerous studies have been conducted to enhance the FR of composites based on 2D materials by incorporating different flame retardants (Table 2). For instance, Qu et al. [84] demonstrated that blending black phosphorus (BP) with ruthenium sulfonate ligand (RuL_3_) can simultaneously enhance the TC and FR of ECs (Figure 10a). RuL_3_@BP/EP nanocomposites are easily able to achieve the vertical burning (UL-94) V-0 rating, and their limiting oxygen index (LOI) value of 26.72% when RuL_3_@BP is added to the resin at 3 wt%. while the total heat release (THR) lowers by 35.22%, the peak heat release rate (pHRR) decreases by 62.21%. Additionally, the high TC of the nanocomposites is notable at 0.376 W/(m·K), which represents improvements of 52.23% and 65.64%, respectively, above BP/EC (0.247 W/(m·K)) and EP (0.227 W/(m·K)). Wang et al. [87] reported that zinc hydroxystannate (ZHS) modified MXene (Ti_3_C_2_T_x_) can improve the thermal management capacity and safety of EP. When the ZHS@MXene is added to epoxy at 2 wt%, the TC (0.87 W/(m·K)) is enhanced by 328% compared to the EP. The pHRR and THR decrease by 54.91% and 58.74%, respectively.

Even though 2D materials possess outstanding TC and are potential candidates for enhancing the TC of ECs, they suffer from severe agglomeration behavior and weak interfacial interaction with the matrix due to their poor interfacial compatibility. This incompatibility issue prevents the formation of TC paths and weakens the barrier effect, which ultimately leads to decreased TC and FR [88]. Consequently, the synthesis of 2D materials that exhibit excellent dispersion and FR simultaneously is crucial for fabricating high TC and FR ECs. Chang et al. [89] synthesized a composite of MoS_2_, silver nanowires (AgNWs), and 3D reduced graphene oxide (RGO) aerogel micro-particles (AgNW-RGO@MoS_2_), which demonstrated remarkable dispersibility in EP as depicted in Figure 10b. The composite containing 4.0 vol% of AgNW-RGO@MoS_2_ showed outstanding TC enhancement up to 420%. Additionally, the pHRR and THR were reduced by 61.1% and 58.8%, respectively, when compared to EP. In the case of the AgNW-RGO@MoS_2_/EP composite, the pre-constructed 3D AgNW-RGO@MoS_2_ framework effectively prevented filler aggregation. When introduced into the EP, a highly efficient TC microregion was established, and effective heat transfer was achieved. In addition, the AgNW-RGO@MoS_2_ prevented mass (combustible gases or smoke particles) from being exchanged between the heat source and substrate, which leads to superior flame-retardant properties.

Recently, grafting flame-retardant elements into the epoxy matrix in a synergic approach to enhance FR efficiency has been regarded as one of the most effective and eco-friendly methods for flame retardants. Yang et al. [90] developed an intrinsic flame-retardant epoxy (DER/MEP-DDM) using vanillin as the base material, as depicted in Figure 10c. This formulation significantly improved the fire resistance of the epoxy. To further enhance its properties, graphene aerogel (GA) was incorporated into the DER/MEP-DDM matrix, resulting in a composite material with excellent TC and FR performance, even with a low loading of 0.5 wt% GA. The FR performance was attributed to the quenching effect of free radicals, the dilution effect of nonflammable gases, and the formation of a protective char layer that hindered heat and oxygen diffusion. Meanwhile, the superior TC has been attributed to the 3D graphene TC pathways that were integrated into the matrix.

Zhong et al. [91] developed a highly thermal conductive and flame-retardant epoxy material (D-LCER_DP_) by designing and synthesizing a triphenylene-based discotic liquid crystal epoxy (D-LCE) and a co-curing agent (DOPO-POSS, DP) containing P/Si flame retardant elements (Figure 10d). The D-LCE was cured with 4,4′-diamino diphenyl methane (DDM) and DP to produce the D-LCER_DP_ material within a liquid crystal range. When DP was added at a 10 wt% ratio, the resulting D-LCER_DP_ material exhibited excellent intrinsic high TC (in-plane TC of 1.30 W/(m·K) and through-plane TC of 0.34 W/(m·K)) as well as high FR (LOI up to 31.1 and UL-94 up to V-0 level). The high FR was attributed to the low smoke and release volume, low burning rate, and the prevention of melt drops caused by the introduction of P and Si flame retardant elements. The high intrinsic in-plane TC was due to the regular arrangement of the LCERs.

**Figure 10 polymers-15-02818-f010:**
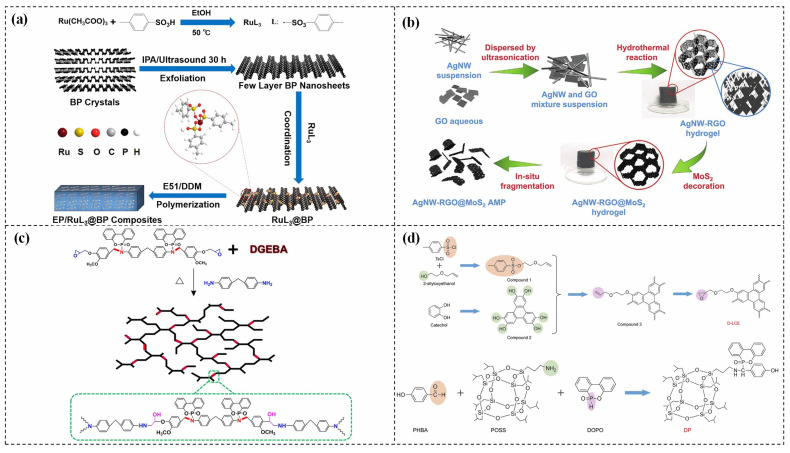
(**a**) Preparation, flame-retardant mechanism, and TC mechanism of RuL_3_@BP/EP composites [84], Copyright 2020. Reproduced with permission from Elsevier Ltd.; (**b**) Preparation of AgNW-RGO@MoS_2_ and flame-retardant mechanism of AgNW-RGO@MoS_2_/EP [89], Copyright 2022. Reproduced with permission from Wiley-VCH GmbH; (**c**) Curing process of DER/MEP-DDM systems and schematic diagram of the FR mechanism of GA/DER/MEP-DDM composites [90], Copyright 2021. Reproduced with permission from American Chemical Society; (**d**) Schematic diagram of synthetic route for D-LCE and flame-retardant co-curing agent DP, and schematic diagram of a flame-retardant mechanism for D-LCER_DP_ [91], Copyright 2021. Reproduced with permission from Wiley-VCH GmbH.

**Table 2 polymers-15-02818-t002:** The key properties of flame retardancy thermal conductivity epoxy composites.

Filler Additives	EP Matrix	TC W/(m·K)	Highlights in Flame Retardancy Properties	Refs.
12.1 vol % ILFR-fBNNSs/epoxy	DGEBF, YDF-170	1.04	PHRR: 42%↓ THR: 37%↓	2018 [15]
7 wt% APP/g- C3N4/epoxy	E-44	1.09	LOI: 30.1% UL-94: V-0	2019 [92]
20 wt% hBN/2 wt% RGO@Ni(OH)2/epoxy	DGEBF, YDF-170	0.66	PHRR: 33.5%↓ THR: 33.8%↓ LOI: 29.2% UL-94: V-1	2020 [88]
3 wt% RuL3@BP/epoxy	E-51	0.376	PHRR: 62.21%↓ THR: 35.22%↓ LOI: 26.72% UL-94: V-0	2020 [84]
0.5 wt% GA/Vanillin-based epoxy	Vanillin-based epoxy (self-made)	0.592	PHRR: 27%↓ THR: 35%↓ LOI: 27.5% UL-94: V-0	2021 [90]
CCA/ 9.6 wt% m-BN aerogel/epoxy	E-44	2.11	PHRR: 56.89%↓ THR: 36.86%↓	2021 [93]
DP-DDM-D-LCE	D-LCE (self-made)	In-plane TC,0.34 Through-plane TC,1.3	LOI: 31.1 % UL-94: V-0	2021 [91]
21.3 wt% 3D BN/MWCNTs/C/epoxy	E-44	1.84	PHRR: 55.4%↓ THR: 36.9%↓	2021 [94]
Functionalized BNNS2/epoxy	DGEBF, YDF-170	0.999 (30 wt% BNNS)	PHRR: 60.9%↓ THR: 35.7%↓ (5 wt% BNNS)	2021 [21]
4 vol% AgNW-RGO@MoS2/epoxy	DGEBF, YDF-170	0.85	PHRR: 61.1%↓ THR: 58.8%↓	2022 [89]
2 wt% ZHS@MXene/epoxy	DGEBA	0.8691	PHRR: 54.91%↓ THR: 58.74%↓	2022 [87]
BNNS-DOPO/epoxy	DGEBA	1.25 (30 wt% BNNS-DOPO)	PHRR: 47.6%↓ THR: 44.7%↓ (20 wt% BNNS-DOPO)	2022 [95]
50 wt% F-BN/PPNV-NH2 /epoxy	DGEBA	2.52	LOI: 30.6% UL-94: V-0	2023 [82]
2 wt% FeHP@GO/epoxy	E-51	0.433	PHRR: 46.2%↓ THR: 23.5%↓ LOI: 33.5% UL-94: V-0	2023 [40]
PDA-BNAO/epoxy	EP JY257	1.192	PHRR: 66.2%↓ THR: 45.5%↓	2023 [96]
3D 12.08 g BN/12.08 g talc/6.04 g APP/epoxy	E-51	3.04	LOI: 37.8% UL-94: V-0	2023 [97]

### 3.4. Electrothermal Epoxy Composites

Electrothermal composites (ETCs) are a crucial area of research in the field of conductive materials. When ETCs are subjected to voltage, their internal conductive filler acts as a heater, and using Joule heating, they convert electricity into heat. The heat generated by ETCs can be calculated using Joule’s law (Q = U^2^t/R), where Q represents heat generation, U is the input voltage, t is time, and R is resistance. This law clarifies that ETCs’ Joule heating performance is affected by both the voltage applied and the resistance of the conductive filler [27]. Additionally, there is a percolation effect between resistivity and the amount of conductive filler present, and the threshold value required for conductive channels to form can be quantitatively determined. In addition to possessing good electrical conductivity, ETCs must also have high TC to facilitate efficient heat transfer and improve the material’s overall thermal stability [98]. This is especially important in applications where the electric current generates heat, as excessive heating can lead to element failure or damage. High TC materials can dissipate heat quickly, reducing the risk of overheating and ensuring consistent and reliable heating performance. EP is a widely studied matrix in the literature and has been utilized in various applications, including deicers, defrosters, and electrothermal actuators.

#### 3.4.1. Deicers, Defrosters, and Foggers

ETCs are widely recognized for their outstanding Joule heating performance, which makes them ideal for anti-icing and de-icing applications in modern transportation and communication industries such as automotive windows, outdoor displays, communication devices, and power supply equipment [99]. To facilitate facile compatibility with structural composites, epoxy matrices are the preferred choice for these applications. To enhance the electrothermal efficiency of these composites, state-of-the-art carbon nanostructures as carbon fibers [100], CNTs [101,102], and graphene derivatives [103,104] have been incorporated into EPs due to their high electrical and TC and excellent mechanical robustness [105].

Redondo et al. [103] present their research on the suitability of GNP/epoxy coatings for anti-icing and de-icing systems. Their study shows that doping the coatings with GNP contents between 8–12 wt% exhibits optimal anti-icing behavior at −15 °C, while higher GNP contents (12 wt%) are necessary for effective de-icing systems at −30 °C. These findings suggest that GNP/epoxy coatings have significant potential for use in critical anti-icing and de-icing applications. Sanchez-Romate et al. [106] further explored GNP/epoxy coatings for de-icing applications and found that an increase in GNP content promotes more efficient resistive heating. However, an increase in GNP content in the coatings leads to a more heterogeneous material with a higher prevalence of porosities, which can result in lower-quality coatings (Figure 11a). Yang et al. [98] have designed and demonstrated a 3D graphene-based aerogel with remarkable anisotropic Joule heating performances when combined with ECs. The composites, which have a 4.7 wt% loading of graphene-based aerogel, exhibit excellent Joule heating properties of 213 °C at a relatively low applied voltage of 5V. Additionally, the graphene-based aerogel/ECs exhibit excellent electrical conductivity of 48.7 S/m and TC of 1.1 W/(m·K) along the parallel direction to the lamellar structure growth, as depicted in Figure 11b.

Farcas et al. [101] investigated the impact of CNTs and graphite additives in the same epoxy matrix on the self-healing capacity of the composite material and its de-icing and ice-prevention capabilities. The study revealed that the CNTs, with their higher aspect ratio, led to greater electrical conductivity of the EP compared to the graphite-filled resin. In contrast, the 2D morphology of graphite resulted in higher TC in the filled EP. Furthermore, the presence of graphite improved the thermal stability of the filled EP, which helped prevent its deformation due to the softening of the EP. These findings highlight the potential of combining CNTs and graphite to achieve high-performance de-icing and ice-prevention composite materials.

MXenes are a novel class of two-dimensional materials that possess unique properties, including high electrical conductivity (~106 S/m) and excellent TC [107]. These properties make MXenes highly attractive for a variety of applications, including ETC. In a recent study by Yang et al. [108], a 3D Ti_3_C_2_T_x_ MXene/EC was synthesized and investigated for its Joule heating performance. The results showed that even at a low voltage of 2 V and 5.1 A current, the composite demonstrated excellent performance, achieving a total power output of 6.1 W/cm^2^ (Figure 11c). This highlights the potential of MXene-based composites for use in high-performance Joule heating applications. Tan et al. [109] constructed an alternating chitosan (CS)/MXene multilayered structure via layer by layer assembly approach. Excellent electrical conductivity (969 S/m), EMI shielding efficacy of 10,650 dB cm/g, and in-plane TC of 6.3 W/(m·K) were obtained due to the successful construction of continuous electrical/thermal conduction channels in the MXene layers. Besides, the multilayered films display distinguished Joule Heating performance under comparatively low voltage, such as upon voltage of 8V, the ice cube completely melts within a short time of 240 s.

**Figure 11 polymers-15-02818-f011:**
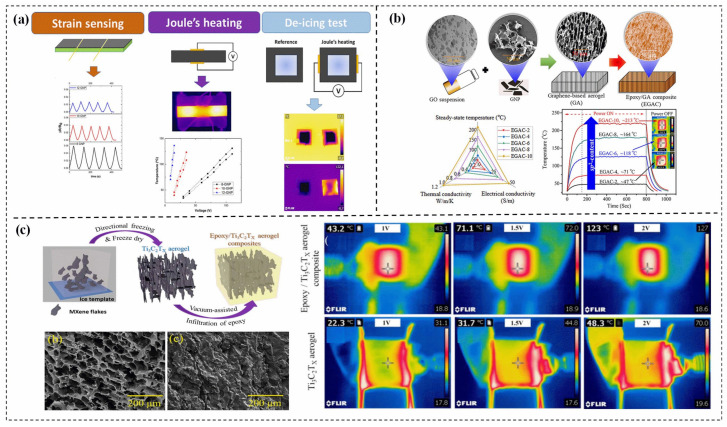
(**a**) Electromechanical response, the temperature reached as a function of applied voltage, and IR images of GNP/epoxy composites [106], Copyright 2022. Reproduced with permission from Elsevier Ltd. (**b**) Preparation schematic, radar plot of longitudinal thermal conductivity, electrical conductivity, and Joule heating performances of graphene-based aerogel composites [98], Copyright 2022. Reproduced with permission from Elsevier Ltd. (**c**) Preparation schematic, SEM, and Steady-stare thermal IR images of Ti_3_C_2_T_x_ MXene/epoxy composites [108] Copyright 2021. Reproduced with permission from IOP Publishing Ltd.

#### 3.4.2. Electrothermal Actuators

Electrothermal actuators are commonly used in microelectromechanical system (MEMS) devices due to their low voltage operation and simple fabrication process. These actuators have been utilized in various applications, including precise-tracking positioning devices, micromirrors, nanoprobes, and manipulators [110]. Zhu et al. [111] developed an artificial dragonfly wing based on a hybrid actuator made of graphene-on-organic film. The reversible mechanical actuation and transparency of graphene provided a foundation for the design of an artificial dragonfly wing that could provide both drives and support the structure. The flapping and bending movement of the wing can be controlled by adjusting the frequency and duration of the applied voltage. Additionally, Slobodian et al. [112] investigated the effect of embedding electro-conductive multiwalled carbon nanotube nano-papers in an epoxy matrix on the release of the frozen actuation force and actuation torque in the resulting composite after being heated over its glass transition temperature. The presence of the nano-paper significantly enhanced the recovery of the actuation stress, increasing it by a factor of two compared to pure epoxy strips. These findings demonstrate the potential of incorporating carbon nanotube nano-papers into composite materials for use in actuation and mechanical support applications.

Electrothermal actuators typically consist of two main components: a conductor, which generates heat via the Joule heating effect, and a polymer that expands upon being heated. Commonly used conductive materials for actuators include aluminum foil and copper due to their high electrical conductivity. However, the rigid nature of aluminum foil can limit the bending displacement achievable. In a recent study, Shirasu et al. [113] designed and fabricated electrothermal bimorph actuators using multi-walled carbon nanotubes (MWCNTs), epoxy, and aluminum foil. These actuators were capable of being activated at a low voltage of 6 V, their bending displacement was 10 mm.

### 3.5. Toughness and Thermally Conductive Epoxy Composites

Toughness is a critical property that measures a material’s ability to resist fracture and deformation under stress. Although EPs exhibit high strength and stiffness, they can be brittle and prone to cracking under mechanical stress, such as impacts [23]. To improve their toughness, various toughening agents can be added to EPs, including nanomaterials, thermoplastic polymers, and rubber particles, as shown in Table 3. These agents work by absorbing energy and hindering crack propagation, making the EP more resistant to fracture [28].

While the addition of TC fillers can significantly increase the thermal conductivity of epoxy resins, it can also create a considerable contact thermal resistance between the fillers and the epoxy resin. These defects can become a bottleneck during heat transport and inevitably decrease the toughness of the epoxy resin. Therefore, achieving the desired thermal conductivity without significantly reducing the toughness of the epoxy resin is a major challenge for both scientific research and industrial applications. Kin et al. [114] introduced ozone/tetraethylenepentamine (TEPA)-functionalized nanodiamonds (NDs) into the epoxy matrix to improve its TC and toughness by enhancing interfacial interactions, as illustrated in Figure 12a. The incorporation of ozone/TEPA-functionalized NDs at a loading of 0.50 wt% significantly enhanced the TC and toughness of the nanocomposites by 34.1% and 121.4%, respectively. The excellent interfacial adhesion between ozone/TEPA-functionalized NDs and the epoxy matrix facilitated the formation of a thermal conductive network, resulting in the enhanced TC of the nanocomposites. Moreover, the ozone/TEPA-functionalized NDs composite exhibited higher fracture resistance due to its ability to absorb high energy and strong interfacial adhesion with the epoxy matrix.

Chatterjee et al. [115] demonstrated that the incorporation of 2 wt% graphene nanoplatelets (GnPs) with a diameter of 25 μm significantly improved the fracture toughness and TC of EP by 60% and 36%, respectively. GnPs support crack deflection and crack bridging mechanisms to improve toughness. Similarly, Liu et al. [116] developed an EC with a high through-plane TC of 20 W/(m·K) by impregnating a lamellar-structured graphene aerogel with vertically aligned GnPs into the epoxy matrix. The outstanding thermal conductivity of the composite was attributed to the high thermal conductivity of GnPs and the nacre-like structure of the graphene aerogel. However, the fracture toughness of GnPs/epoxy increased and then decreased with increasing GnPs content, which arose from the interfacial defects between the GnPs and epoxy resin.

Wang et al. [117] introduced polyether sulfone (PES) into the GnPs/epoxy system to fabricate TC GnPs/PES-epoxy composites with high toughness. The fracture toughness was further enhanced by about 29.5% relative to the PES-modified epoxy at the same GnPs concentration. Thermoplastics such as PES could enhance the mechanical properties of epoxy resin through the formation of a varied phase structure via reaction-induced phase separation. Luo et al. [118] improved both the TC and mechanical properties of ECs by incorporating PES and silver nanoparticles functionalized boron nitride nanoplates (AgBNs). The TC of AgBNs/PES-EP achieved a value of 0.54 W/(m·K) at a loading of 10 wt% AgBNs. The higher fracture toughness of AgBNs/PES-EP than AgBNs/EP has arisen from the formation of the distinctive PES and EP phase structure, further impeded by rapid crack expansion. In addition, the fracture mode of AgBNs/PES-EP gradually changed from brittle to ductile.

The toughening effect of soft rubber was proved to be significant [119,120], and major toughening mechanisms involve rubber phase debonding/cavitation, localized shear banding of the matrix as well as rubber particle bridging/tear. Wang et al. [28] successfully prepared GnPs/ carboxyl-terminated butadiene acrylonitrile (CTBN)/epoxy ternary composites with significantly enhanced toughness (108%) and thermal conductivity (145%) by adding 3 wt% GnP-5 and 10 wt% CTBN to EP. as depicted in Figure 12b. The toughening mechanisms in CTBN/epoxy were found to be rubber cavitation and deformation, and shear bands formation. Furthermore, the incorporation of GnPs further enhanced the fracture toughness of the rubber-modified epoxy, which was attributed to the larger crack area induced by crack deflection, matrix shear banding, layer breakage, and separation/delamination of GnPs layers in the ternary composite.

**Figure 12 polymers-15-02818-f012:**
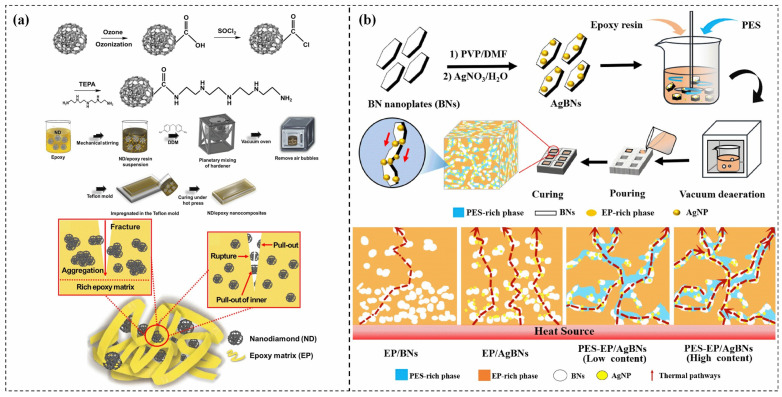
(**a**) Schematic illustration of the preparation process and toughness mechanism of ozone/TEPA-ND composites [114], Copyright 2020. Reproduced with permission from Elsevier Ltd.; (**b**) Schematic illustration of the preparation process and thermal conduction mechanism of BNs/EP, AgBNs/EP, and AgBNs/PES-EP [118], Copyright 2023. Reproduced with permission from Elsevier Ltd.

**Table 3 polymers-15-02818-t003:** The key properties of oughness and thermally conductive epoxy composites.

Toughening Agents		EP Matrix	TC W/(m·K)	K_IC_ MPa·m^1/2^	Refs.
Nanomaterials	0.5 wt% Ozone/TEPA-functionalized ND/epoxy	YD-128 DGEBA	0.4	15	[114]
	2 wt% 25 μm GnP/epoxy	EPIKOTE 828LVEL	0.70	0.91	[115]
	2 wt% 3nm in thickness GnP/epoxy	WSR 618 DGEBA	0.23	1.75	[23]
	20 wt% NGFs/BFs/epoxy	YD-128 DGEBA	0.633	90.56	[121]
	2.3 vol% Graphene aerogels/epoxy	DGEBA	20 (through-plane)	0.70	[116]
Thermoplastic	3 wt% GnPs/PES-EP	DGEBA, Epon 828	0.37	1.36	[117]
	10 wt% AgBNs/PES-EP	Epoxy resin 2400	0.54	2.21	[118]
Rubber	3 wt% GnPs/ 10 wt% CTBN-EP	DGEBA, Epon 828	0.49	1.58	[28]

## 4. Summary and Challenges

The performance of single-function TC ECs and devices has been greatly improved after decades of development. Taking into account actual applications, materials designed to increase the TC value are not able to satisfy the increasing demand. In certain cases, TC is not the only acceptance criterion, which accelerates the development of multifunctional TC ECs. This review highlights the preparation strategies, mechanisms, and applications of five kinds of mainstream multifunctional TC ECs innovated in the past few years towards sustainable, multifunctional, and high-performance thermosetting materials. Also, there are many commonalities in the preparation strategies between multifunctional TC ECs by carbon-based fillers. Thus, this review will be able to provide essential and comprehensive information on how to develop multifunctional (electromagnetic shielding properties, flame retardant properties, electrical and thermal properties, toughening) TC ECs as well as motivate more innovations in this community, but despite several achievements, challenges remain.

In thermally conductive epoxy vitrimers, the presence of a segregated conductive network provides isotropic TC but hinders the mobility of polymer chains. This trade-off between enhanced TC and hindered topological rearrangement poses a significant challenge. The use of high-loading fillers may impact the topology rearrangement while low-loading fillers would not be enough to TC. As such, there is a pressing need to balance TC and topo-logical rearrangement of polymer chains and develop advanced epoxy vitrimers with high TC at minimal filler loading;There has been considerable advancement in the field of thermal conductive EMI shielding composites. Nevertheless, the production of these composites often necessitates multiple processing stages, such as freeze-drying, and vacuum-filtration self-assembly, among others. The intricacy of these processing steps may enhance the likelihood of defects and impurities in the final product, ultimately compromising its properties;To achieve the desired level of FR, a substantial amount of flame retardant must be incorporated into the epoxy matrix. However, high levels of filler inclusion may lead to brittle composites and pose additional processing challenges. Although the incorporation of flame-retardant elements into the epoxy matrix can partially or completely address these issues, it also has defects such as complicated procedures, which seriously impede its practical application. Therefore, further research is required to improve the FR and TC properties of these composites;A thorough evaluation of electrothermal ECs is essential to determine their performance and reliability. This entails assessing their heating efficiency, durability, and resistance to environmental factors such as moisture. To ensure accurate and reproducible results, it is necessary to establish and standardize testing protocols. Moreover, the integration of electrothermal ECs into specific applications can be a daunting task. For instance, the design of de-icing and defogging systems must account for various factors, including airflow, temperature gradients, and power requirements;Enhancing the toughness of ECs often entails incorporating elastomers or other flexible polymers, which can compromise TC by creating an insulating effect. Conversely, the incorporation of fillers or particles to improve TC may decrease the toughness of the composite. As a result, achieving both toughness and TC in ECs poses a challenge that necessitates the meticulous selection of fillers or additives, optimization of processing conditions, and a comprehensive understanding of the structure-property relationships of the final composites.

In summary, the rapid and essential development of TC ECs has witnessed a transformation from single-functionality to multifunctionality in the last few years. We believe that the widespread application of electronic, electrical, and energy fields will motivate more possibilities and promote better human life experiences in the future through multifunctional TC ECs. However, current multifunctional thermal conductive composites are still in their infancy, and they suffer from some trade-offs between their inherent nature and multiple functions, which hinder their ability to achieve overall performance in one system. Therefore, it is essential to carry out further cross-disciplinary research involving materials science, engineering, chemistry, physics, and electronics to achieve advanced multifunctional TC ECs.

## Figures and Tables

**Figure 2 polymers-15-02818-f002:**
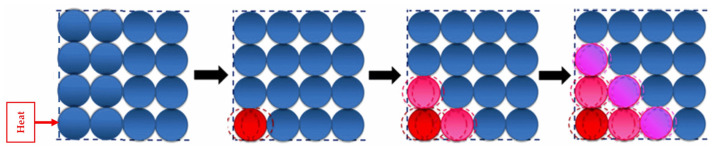
Illustration of the temperature gradient in a crystalline material [32]. Copyright 2016. Reproduced with permission from Elsevier Ltd.

**Figure 3 polymers-15-02818-f003:**
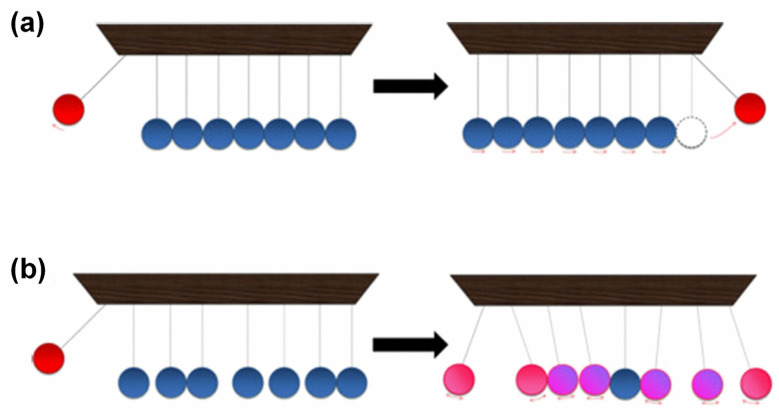
Schematic comparison by the Newton pendulum of thermal conductivity in (**a**) crystalline and (**b**) amorphous materials [32]. Copyright 2016. Reproduced with permission from Elsevier Ltd.

**Figure 4 polymers-15-02818-f004:**
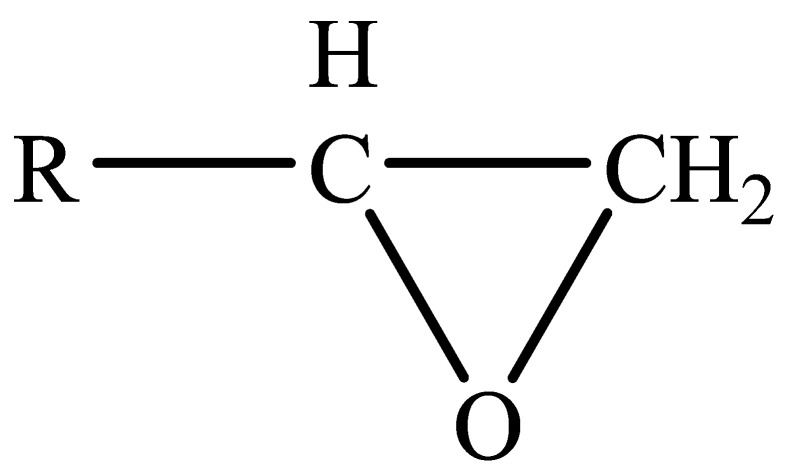
Chemical structure of the epoxy ring.

**Figure 5 polymers-15-02818-f005:**
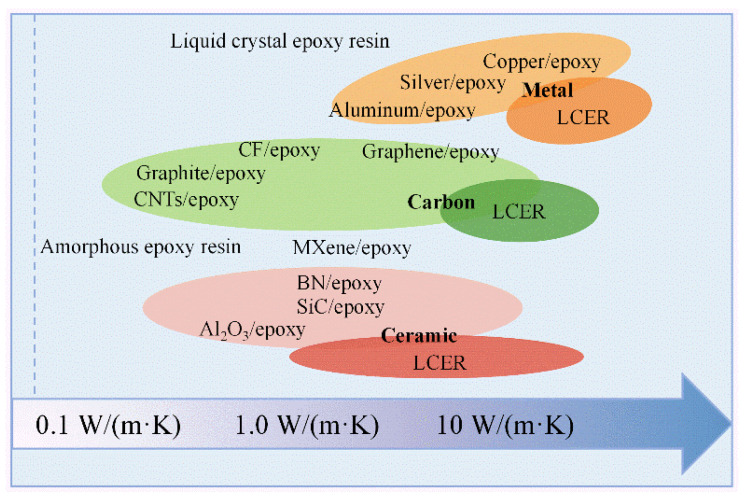
The summary of ECs with different TC values.

**Figure 6 polymers-15-02818-f006:**
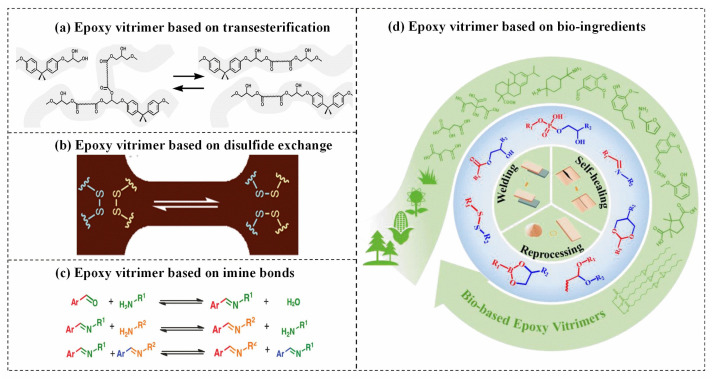
(**a**) Illustration of transesterification in the hydroxyl-ester network [59], Copyright 2011. Reproduced with permission from AAAS; (**b**) Exchange of disulfides in the epoxy backbone enables reprocessing [61] Copyright 2018. Reproduced with permission from American Chemical Society; (**c**) Three equilibrium reactions of imines [62], Copyright 2012. Reproduced with permission from Royal Society of Chemistry; (**d**) Synthetic of eugenol epoxy and epoxy vitrimer, and heat-induced transesterification reactions [63] Copyright 2023. Reproduced with permission from Elsevier Ltd.

**Figure 7 polymers-15-02818-f007:**
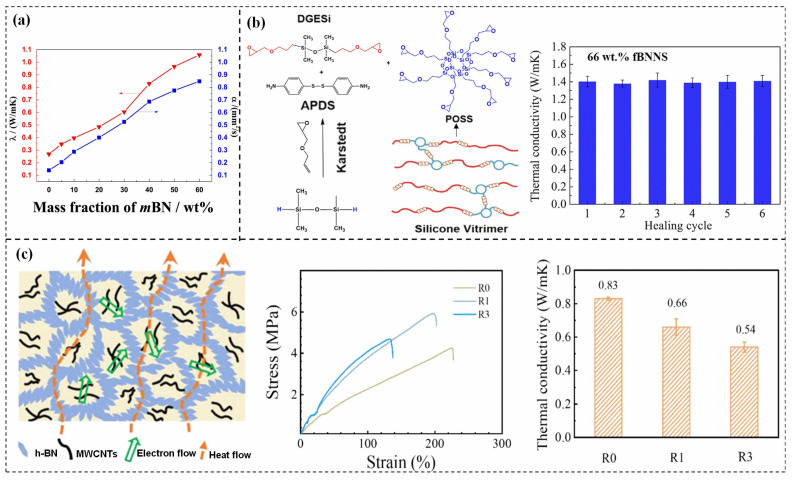
(**a**) TC of the mBN/thiol-epoxy elastomer composites [64], Copyright 2018. Reproduced with permission from Elsevier Ltd.; (**b**) Chemical structure of silicone vitrimer and the fabrication process of TC composite, and Recovery of the TC of the composites after different healing cycles [26], Copyright 2022. Reproduced with permission from Elsevier Ltd.; (**c**) Schematic illustration of the preparation of s-EV/MWCNTs/h-BN composites, TC of composites, and stress-strain curves of reprocessed composites [65]. Copyright 2022. Reproduced with permission from the American Chemical Society.

**Figure 8 polymers-15-02818-f008:**
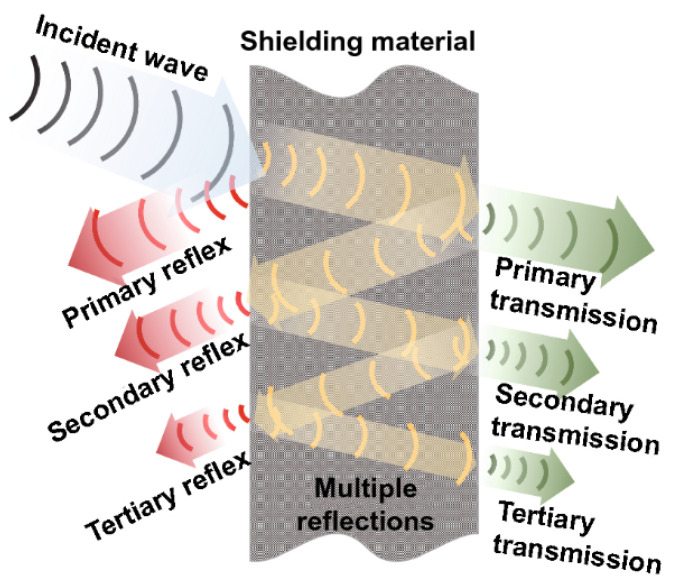
Schematic diagram of EMI shielding mechanism based on transmission line theory [68], Copyright 2021. Reproduced with permission from The Author(s).

**Figure 9 polymers-15-02818-f009:**
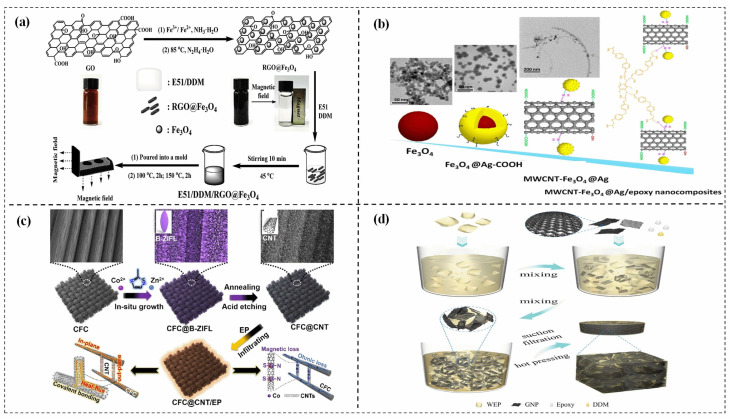
Structures, EMI shielding, and thermal conductivity properties of epoxy composites. (**a**) RGO@ Fe_3_O_4_/epoxy composites [25], Copyright 2019. Reproduced with permission from Elsevier Ltd. (**b**) MWCNT-Fe_3_O_4_@Ag/epoxy composites [74], Copyright 2019. Reproduced with permission from Elsevier Ltd. (**c**) CFC@CNT/epoxy composites [20], Copyright 2022. Reproduced with permission from the American Chemical Society. (**d**) GNP/WEP/epoxy composites [75], Copyright 2022. Reproduced with permission from Elsevier Ltd.

**Table 1 polymers-15-02818-t001:** The TC of commercial EPs.

Epoxy Resins	Monomer	Hardener	TC ValueW/(m·K)	Ref.
Amorphous EPs	DGEBA	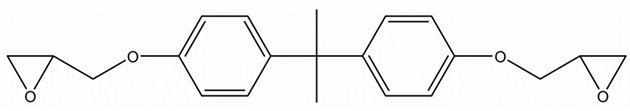	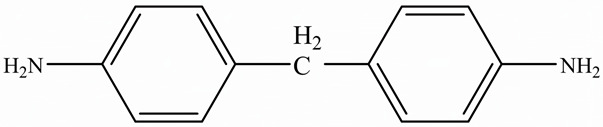	0.19	[40]
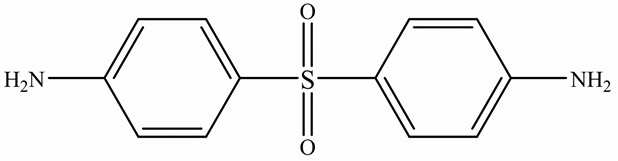	0.20	[41]
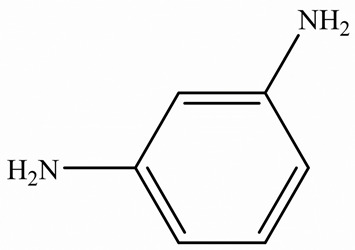	0.20	[42]
DGEBF	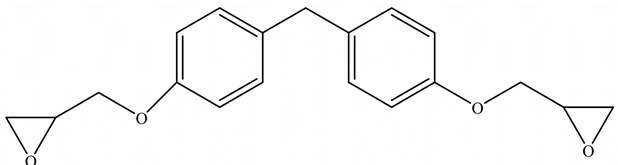	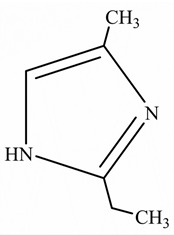	0.201	[21]
LCERs	BPE	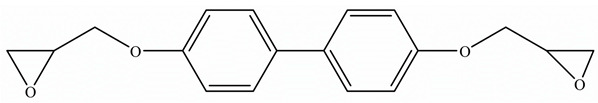	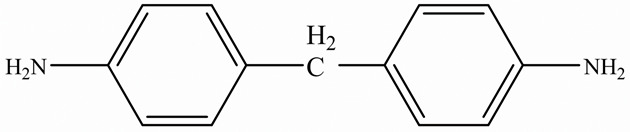	0.3	[38]
TMEn	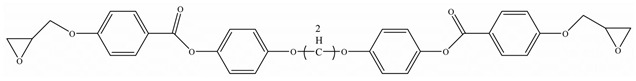	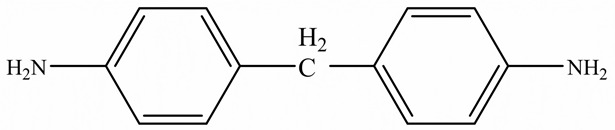	n = 8, 0.85n = 6, 0.89n = 4, 0.96	[38]
LCE1	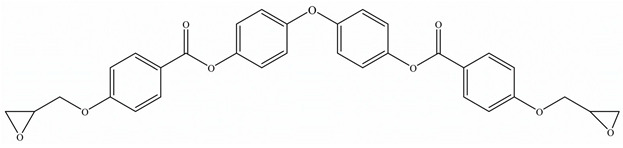	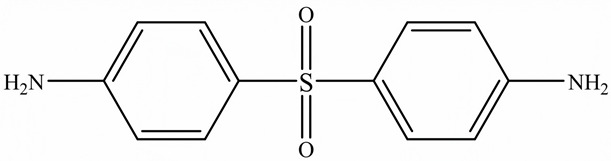	0.292	[43]
LCE-TA	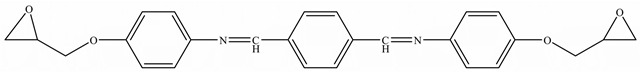	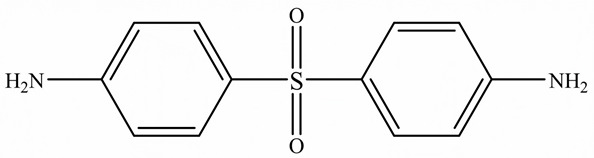	0.45	[44]

## Data Availability

Data is contained within the article.

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
