# Peer review of "Recent Progress on Multifunctional Thermally Conductive Epoxy Composite"

_polymers, 2023, doi:10.3390/polym15132818_

Round 1

Reviewer 1 Report

The authors of the review article analyzed the publications and made recommendations for the development of multifunctional thermally conductive epoxy composites to meet the growing requirements of the next generation of materials. There are just a few recommendations:

1. I propose to add an analysis of one article to the review:

https://www.tandfonline.com/doi/abs/10.1080/15421406.2015.1137122

2. It would be useful to show correlations between the mechanical and thermophysical properties of thermally conductive epoxy composites. So far, this information is shown only in the form of a description and an difficult for generalization.

3. There are practically no formulas in the article, which is a pity. This does not allow showing the design parameters and criteria that determine the thermal properties of various thermally conductive epoxy composites.

4. Perhaps it was worth showing a table in which to conduct a comparative analysis of the properties of various thermally conductive epoxy composites.

5. It was also worth showing the technological features of creating multifunctional thermally conductive epoxy composite with various pre-set properties.

Author Response

Dear reviewer,

Thank you very much for your comments and suggestions. I send you our answers in the enclosed manuscript. In it, we include the response for the second reviewer in order to contestualize all the changes. 

Reviewer 2 Report

The systematic review presented by the authors is devoted to the actual topic - the creation of epoxy polymers with high heat-conducting properties. A small revision of the article is needed on the following issues:

1. The article focuses on the introduction of dispersed fillers of various chemical nature into epoxy resins and their effect on its thermal conductivity. However, reinforcing fillers are also often introduced into epoxy polymers.

2. Currently, heat-conducting epoxy adhesives, sealants, and potting compounds, in particular for microelectronics, are widely used, but this article does not pay attention to this class of epoxy materials.

Author Response

Dear reviewer,

Thank you very much for your comments and suggestions. We send you our answers in the uploaded manuscript. In it, we include the response for the  reviewer 1 in order to contestualize all the changes. 

Thank you very much 

Round 2

Reviewer 1 Report

Accept.